# Designer Ecosystems for the Anthropocene—Deliberately Creating Novel Ecosystems in Cultural Landscapes

Jason Alexandra 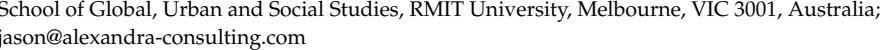

School of Global, Urban and Social Studies, RMIT University, Melbourne, VIC 3001, Australia;
jason@alexandra-consulting.com

**Abstract:** Accepting that nature and culture are intricately co-evolved has profound implications for the ethical, legal, philosophical and pragmatic dimensions of social and environmental policy. The way we think about nature affects how we understand and manage ecosystems. While the ideals of preserving wilderness and conserving ecosystems have motivated much conservation effort to date, achieving these ideals may not be feasible under Anthropocene conditions unless communities accept custodial responsibilities for landscapes and other species. This paper's origins are in the author's work with the Martuwarra Fitzroy River Council representing Indigenous traditional owners in Australia's Kimberley region. These landscapes, shaped by 60,000 years of human occupation, interweave knowledge, laws and governance regimes, and material and spiritual connections with country. This interweaving offers insights into options for dealing with humanity's complex sustainability challenges. The paper also draws on the literature about cultural landscapes, ecological design, agroecology and permaculture to explore options for applying ecological design as a planning and problem-solving framework. The paper concludes that design-based approaches offer significant opportunities for using ecological science to integrate conservation and production in agricultural landscapes in ways that can meet human needs while also conserving biodiversity under climate change.

**Keywords:** designer ecosystems; ecological restoration; environmental policy; Anthropocene; agroecology; permaculture; cultural landscapes; assemblages

## 1. Introduction

Different cultures have fundamentally different relationships with animals and the environment due to their foundational beliefs and worldviews [1]. These shape nature–culture relationships and underpin the way ecosystems are understood and managed [1,2]. Most contemporary ecosystem management models have their origins in the European enlightenment, embedding the Cartesian duality that separates humans from nature [3]. The dominant, instrumental, administratively rational models perpetuate this dualism, separating the knowing and governing of nature into science and policy domains and outsourcing environmental management to professional experts [1]. These models were promoted globally via colonial regimes and their attempts to "civilise" the World's "wilderness areas" and "wild" Indigenous peoples [3]. The results have profound implications for the Earth's diverse peoples and landscapes and the way ecosystems are managed [3]. The administrative and scientific rationalism that underpins modern environmental management grew to dominance under conditions of relative climatic stability [4]. However, the pressures of a new climate regime mean we need to critically examine and reconceive our relationships with the Earth, the biosphere and other species [4]. This critical examination should extend to the foundational concepts and normative values underpinning nature–culture relationships, which are often taken as givens.

Taking up the challenge of this critical examination, in this paper, I explore some of the conceptual issues involved in working to achieve conservation outcomes in landscapes

dominated by human activity, such as agriculture. In particular, I explore why ecological design [5] provides a useful framework for ecosystem management. Throughout this paper, the term landscape is used to define the co-evolved ecosystems where nature and culture are blended, negotiated and merged into socio-natural assemblages [6]. These assemblages are often referred to as cultural landscapes, with approaches to their management drawing on advances in the social and natural sciences [7,8].

The paper's origins are in the author's work with the Martuwarra Fitzroy River Council, which represents Indigenous people in Australia's Kimberley region. These traditional owners find abhorrent the widely promoted idea of wilderness. Instead, they seek greater recognition of their relational models of cultural landscape management. For a clear articulation of these concerns, see Fletcher et al. [3]. This paper goes beyond these concerns about wilderness, exploring options for meeting complex sustainability challenges by drawing on literature about cultural landscapes, ecological design, agroecology and permaculture. The aim of this paper is to explore models of landscape management that recognise the intricate cultural relationships that evolved historically, but which may also provide insights into the complex imperatives and dynamic conditions of the Anthropocene, with its compounding risks and uncertainties [4,5].

The paper is structured as follows:

Following this introduction, the second section explores how Anthropogenic change caused most of the World's terrestrial ecosystems to become anthromes or anthroscapes.

The third section examines lessons from Australia's co-evolved cultural landscapes. While the fourth explores why explicit recognition of the cultural landscapes is needed to guide ecosystem management.

The fifth section sketches out why understanding regional ecosystems as co-evolved assemblages, blending nature and culture, is helpful for reforming models of landscape management in the Anthropocene.

The sixth section explores designer ecology and why ecosystem management in the Anthropocene needs new underpinning logics to guide the transformation of human-dominated landscapes. It explores ecosystem design using ideas about permaculture and agroecology as novel, designed ecosystems.

The final section, before concluding, explores the rationale for further developing the discipline and practice of designer ecology. It explores the prospects for applying ecological science in informing design, planning and management of landscapes at scales from gardens and farms to entire regions.

## 2. Anthromes, Anthroscapes and Cultural Landscapes

The accelerating rates of change occurring to the planet in the Anthropocene [9] build on twelve thousand years of deforestation and land-use change since the advent of agriculture [10]. Through pursuing food and fibre production using agriculture, humans have altered about two-thirds of the World's ice free, terrestrial ecosystems, forming the extensively modified landscapes or anthropogenic biomes that Ellis [11] defines as anthromes. Human-dominated regions that inevitably integrate cultural and natural processes [8,12] are also known as anthroscapes [13] or cultural landscapes [7]. These, co-evolved and productive landscapes are widespread and diverse, sustaining communities from the deserts of Australia [3] to the rain-sodden Lake District of England [8].

Agricultural development simplified many landscapes, resulting in the destruction and degradation of habitats, particularly in those areas best suited to large-scale industrial farming [8,11,14,15]. While not the intent, many species have declined in extent and numbers and ecosystems functions have changed, resulting in large-scale land, water and biodiversity conservation problems [15]. While agriculture has had negative consequences for many species, it is worth noting that some species—known as increasers, such as crop plants, weeds, crows and rodents—have benefited, increasing in number are distribution [16].

Overturning land degradation problems and achieving biodiversity conservation requires the application of principles and guidelines based on ecological science, across multiple scales that range from the local to the trans-national [17]. These principles need refining, for example, through application in the large-scale reafforestation and land restoration programs, such as those proposed in Sub-Saharan Africa [18]. While many policies and programs aim to improve conservation outcomes and reduce the negative impacts of land use, these are rarely commensurate with the scale of the problems. A lack of sustained effort in mobilising governments, industries and communities in transformational landscape-scale conservation initiatives leaves many peopled and production landscapes highly degraded [19]. More fundamentally, there is generally a deep unwillingness to deal with the political, cultural and economic drivers of environmental degradation [20].

Overcoming global-scale problems, such as land degradation and biodiversity loss, requires a multitude of solutions that recognise the cultural and economic dimensions of the people and politics involved in land management [19,21]. Effective nature conservation must involve the people who manage landscapes and cannot only be left to national parks and reserves [19] because nature persists in and through cities [22] and other anthroscapes [15] such as agricultural and pastoral lands [6]. Conservation across all land tenures, beyond traditional parks and nature reserves, requires institutional, policy and practical reforms that mobilise communities and political leaders [19]. These co-evolved cultural landscapes need the care and custodianship provided by communities with their intricate understanding, histories, beliefs, language, experience and governance regimes [3,6,8]. When functional, these normative and cultural frameworks provide capacity for determining that landscapes are managed and governed and can limit the exploitation that drives degradation.

The governance and management processes that shape and reshape landscapes can be seen as a form of ecosystem design [5]. If the shaping and reshaping of landscapes are processes of design, then let us acknowledge humans as the custodians of designer ecosystems. Before elaborating on designer ecosystems, the following section outlines the relevance of Australia's co-evolved cultural landscapes to this idea.

### 3. Learning from Australia's Landscapes Co-Evolved Cultural Landscapes

Australia's co-evolved cultural landscapes provide important lessons for thinking and managing ecosystems in the Anthropocene. There is now a general acceptance that thousands of generations of human use and occupation shaped Australian ecosystems, over periods that spanned several major climatic transitions [23]. Australia's landscapes co-evolved through the iterative and skilful management of Indigenous peoples, including their deliberate and systematic use of fire in ways that determine landscape patterns, species communities and vegetation features [24,25]. In addition, Indigenous peoples extensively modified rivers, wetlands and waterways to form fish traps and freshwater aquaculture systems [26,27] and translocated numerous plant species, including through ceremonial gifting practice [28]. Australia's Indigenous peoples also have deep totemic connections to places and animals [1] and custodial responsibilities for the care of land, water, plants and animals, through first, or natural law, which codified responsibilities for stewarding environmental resources [3,24,29].

The studies outlined above portray cultural governance practices that shaped and nurtured landscapes—including economically, culturally and symbolically important plant and animal species. The attempted silencing and erasure of the evidence of this ecosystem stewardship is part of settler Australia's colonial dispossession of Indigenous peoples that disrupted the practice of natural law and ecosystem stewardship [29,30]. Promoting ideas about preserving "natural" landscapes or "wilderness" areas, free of human influences, reinforces this colonial lineage, furthering the dispossession [31] through what Fletcher et al. [3] call the shackles of wilderness. Claims that only endemic species should be replanted are underpinned by these naturalistic values [32]. However, these nativist approaches ignore the deliberate translocation of plant and animal species by Indigenous

peoples over millennia [27,28]. Despite the attempted silencing, evidence of the continuing heritage values of these cultural landscapes is compelling [23,33], and the continent's profound human history challenges the ideal of "wilderness" based on notions of "natural" ecosystems free of human influence [3,31]. Further, it challenges "wilderness" concepts, which perpetuate the nature–culture dichotomy [3].

In contrast to wilderness, the term country is all encompassing, covering a people's territory, and their relationships to land, water, animal and vegetation resources. It is increasingly commonly used in Australia, to denote the holistic concept of country used by Indigenous Australians [3]. The idea of caring for country is central to many contemporary Australian interactions with the landscape, including under the broad banner of Landcare [19]. For several decades, Australia's governments have actively supported communities' care for the land through policies and programs known generically as Landcare [19]. Landcare programs empower local collaborations in agricultural, coastal and urban areas enabling social, technological and environmental innovations [19]. While Landcare has supported contemporary explorations about living well in this continent, the practice of communities caring for country is at least 60,000 years old and involved laws and practices that stewarded environmental resources [3,24,29,33,34]. Landcare brings together this timeless ethos while fostering innovative and regenerative agriculture and landscape management that integrates the needs of humans and other species [19]. Landcare legitimises diverse relationships with country, promoting creative explorations, innovative practices and new enterprises [19], the need for which is more pressing than ever, given the challenges of climate adaptation [35].

Country is a broad concept that includes the waters and waterways, which have deep cultural, material and spiritual significance [33]. Rivers and their floodplains, not only have high cultural heritage values but also have high biodiversity conservation and material values. Water is central to Indigenous peoples' materiality, sociality and spirituality with increasing claims for the right to govern rivers in ways that recognise water as more than a simple natural resource available for exploitation [33,34,36]. Instead, there are calls to legally recognise rivers as environmental resources and ancestral beings, with cultural identities and legal rights [33].

The critical lesson from examining Australia's cultural landscapes is that landscapes and their components, such as plants, animals and waters, need recognition as co-evolving with people, who have distinct relationships and responsibilities to a territory. In these cultural–natural assemblages, cultural knowledge, laws and governance regimes and material and spiritual connections are intrinsically interwoven with how ecosystems are understood and governed [33]. Recognising these relational dimensions can enable theories and practices of environmental management that better integrate humans and nature. This integration changes the way we understand and govern human–ecosystem relationships and how we define and use natural resources [1,3,33].

## 4. Ecological Restoration under Climate Change?

In this section, I ask whether ecological restoration is relevant under Anthropocene conditions, including climate change. Restoration ecology initially focused on restoring a pre-existing assemblage of species that occupied a site [37,38]. This form of restoration can drive backwards-looking approaches: by definition, the act of restoring implies seeking to re-establish a prior state [37]. Ross et al. [5] outline how recreating past ecosystems was the driving idea behind the formation of the Society for Ecological Restoration, which initially defined ecological restoration as "the process of intentionally altering a site to establish a defined, indigenous, historic ecosystem. The goal of this process is to emulate the structure, function, diversity, and dynamics of the specified ecosystem." The society subsequently altered its purpose, adopting the definition that "ecological restoration is the process of assisting the recovery of an ecosystem that has been degraded, damaged, or destroyed." This conceptual shift acknowledges the impossibility of "recreating historic ecosystems in a world dominated by novel species interactions in a historical climate, biogeochemical, and

hydrological regimes" and that, therefore, global-scale Anthropogenic influences, including climate change, may make some restoration goals unachievable [5].

Ecosystem management in the face of these Anthropocene forces demands flexible and adaptive approaches that can handle the complex multi-scaled and non-linear feedbacks between social, ecological and climatic systems [5]. Hobbs and Harris [38] claim that setting realistic and clear objectives for ecosystem restoration projects is necessary, particularly when working to recover degraded ecosystems (or particular functions of degraded ecosystems).

However, setting clear and feasible objectives may become increasingly difficult due to compounding Anthropogenic drivers of change. These changes can make static conservation paradigms and stationary hydrological models inaccurate or redundant guides to environmental management [5]. These complexities and uncertainties mean that past ways of understanding the world have less utility, and new post-natural or post-normal paradigms are needed [39]. Three examples support the argument that post-natural models of ecosystem management are needed.

Firstly, water resource management and aquatic ecosystem conservation must adjust to the "death of stationarity" which undermines the foundations of hydrology, making past ways of knowing less reliable [40], especially in large river basins affected by climate change [41]. Secondly, pre-development benchmarks used for planning the conservation estate are challenged by climate change altering ecosystems and the distribution of species [42]. Thirdly, climate and land use change is altering wildfire dynamics. The increasing scale, intensity and impacts of wildfires in many parts of the world have profound implications for disaster management, conservation and land-use planning [43]. Many of the world's forests and woodlands appear to be at fire-driven "tipping points" that could change species distribution, vegetation types and ecosystem dynamics [44]. The intensification of wildfires is due to multiple, compounding human causal agents including climate, land use and ecosystem change [44], indicating caution is needed about simplistically and deterministically reducing the future to climate [45].

Given these compounding change drivers, linear projections or static views of "nature", "natural" systems or "natural" regimes for fire, water or biodiversity have limited utility. Theoretical models that can accommodate the increasingly dynamic nature of ecosystems are needed and governance regimes of socio-ecological systems need to recognise the potential for profound shifts to radically altered states [46]. Evidence of this potential for radical shifts in ecosystems can be found in many parts of Australia, where colonial-settler disruptions have fundamentally and dramatically altered much of the country's terrestrial and aquatic ecosystems [47]. Risks of further major shift are intensifying due to compounding anthropogenic influences, including climate change, [5,39], which is profoundly affecting major river systems [41,48]. The scale of change occurring in Australia's Murray–Darling Basin is leading to fundamental questions about ecological "restoration" as an explicit policy or implicit normative goal [48,49]. For example, Harris [49] argues that many restoration goals are unrealistic and that more critical examinations are needed of the failures to restore riverine ecosystems, despite the decades of work and billions of dollars expended [49].

## 5. Disturbance or Co-Creation—Beyond Ecosystem Restoration

This section briefly explores some ecological concepts that may be useful in the post-normal Anthropocene conditions. However, it is important to note that ecology is a diverse discipline that uses multiple techniques, theories and paradigms [50]. These theoretical frameworks and tools are evolving and complementary, serving different ends in different situations, locations and operations [50].

Recombinant ecology focuses on studying novel assemblages of species, including, for example, invasive species [37]. Recombinant ecologies are designed, hybrid and unintended novel combinations of biotic and non-biotic elements that form into new species assemblages. Anthropogenic impacts tend to enhance ecosystem novelty, including for

example, through climatic change shifting species beyond their historical range [42] and by the globalised flows of genetic materials, resources, goods and people.

Embracing ideas about novel and recombinant ecosystems has significant implications for ecosystem management [38]. Perturbation and disturbance are concepts commonly used in ecology to explain the heterogeneity of ecosystems. These disturbances are often driven by humans or by other processes, such as climate variability (e.g., droughts and floods), by geological processes such as volcanism or by the spread of invasive species, disrupting established ecosystem relationships.

Higgs [51] offers a definitional framework for ecosystems based on degrees of disturbance and the resources, efforts and interventions needed for effective restoration. They argue that a spectrum exists from intact ecosystems, restorable hybrids to the novel, recombinant and designer systems, and that the intensity of management interventions increases as systems move away from stable intact systems toward designed ecosystems. Under this definitional framework, farms and urban areas are hybrids, novel or designer ecosystems because they have been modified beyond cost-effective restoration. However, some restoration programs are also management intensive, depending on project goals and the specific interventions needed, such as predator-proof fencing [52] or large-scale re-engineering of floodplains for environmental water application [53]. While an understanding of "undisturbed" or reference ecosystems may be useful, in many situations, deriving planning objectives from historically assumed "natural" or pre-development benchmarks can be limiting in a world of dynamic change [54] and may lead to the establishment of societal objectives for ecosystem management that have little regard for the costs or the feasibility of success [49]. For example, many river restoration programs suffer from unachievable aspirations and unrealistic goals [49].

Disturbance concepts also underpin definitions of wilderness as areas relatively undisturbed by human activities [55]. However, somewhat ironically, the very act of defining an area as a wilderness, and therefore, worthy of conservation, is a human activity. This is because all decisions to define and "preserve" wilderness are socially produced, reflecting the values, worldviews and governance regimes that determine land use and tenure allocations [3]. These governance regimes reinforce and refract socially constructed values about what are legitimate human interactions with landscapes, limiting other, more diverse ways of knowing and being in the world [3,31]. More importantly, the idea that humans and human activities are disturbances to ecosystems embeds a particular epistemology, reinforcing a binary perspective of human cultures and their interactions with nature [3,55]. Further, there is great productive potential of working with local peoples in the custodianship and co-creation of cultural landscapes, which have multiple and interlinked values, including community resilience and species conservation [3,8,12]. The realisation of this potential requires acceptance of the co-evolution of cultural landscapes and that human influences can be custodial.

Disturbance or degrees of human modification provide one possible classification framework [51]. However, there are others, which do not emphasise human activity as disturbance, but more as the gentle steering and iterative shaping of co-evolved ecosystems [56]. The section following explores this kind of steering through agriculture and the human ecology of landscapes, defining these as designer ecosystems.

## 6. Agriculture as Novel, Designer Ecosystems—Agroecology and Permaculture

Designer ecology focuses on applying ecological knowledge to meet specific objectives, achieving goals through managing ecological functions, rather than an inherent concern with questions of "nativeness" [5]. Both agroecology and permaculture adopt ideas about applying ecological knowledge to human-dominated landscapes. The focus of both is on using management interventions and design-driven approaches to achieve specific defined purposes, such as habitat enhancement or more sustainable human habitation or food production.

Agroecology systems use ecological design principles and approaches to create biologically diverse and productive agricultural ecosystems, which can satisfy the needs of humans and non-human others [57]. Agroecology studies initially focused on understanding farming systems that enhance both agricultural productivity and biological diversity [57]. However, agroecology has broadened to become an assemblage of theory, practice and social movements advocating ecosystem resilience and social and environmental justice [58]. Scholars such as Rosset and Martínez-Torres [58] and Vanhulst and Beling [59] recognise the socio-political dimensions of agroecology, including how it gives expression to the political philosophy—*Buen Vivir*—which celebrates the interdependence of society and nature. This celebration can also be seen in the urban farm movement, which promotes community food initiatives [60].

Permaculture and agroecology are closely aligned in theory and practice, with both advocating that humans can actively create recombinant ecosystems that meet their needs and those of other species [57,61]. Agroecology and permaculture systems combine diverse species from numerous geographic and cultural origins by cultivating a range of domesticated and non-endemic species, along with those that volunteer from the local environment, e..g, weeds, songbirds or insects. Understanding and working with local cultural and ecological resources is fundamental to how these systems function and evolve [57].

Agroecology proponents advocate integrated systems of food production and human habitation, offering solutions to the problems of large-scale, commoditised and industrialised agriculture [58]. The UN Food and Agriculture Organisation [62] actively promotes agroecology as a context-specific, designed approach to optimising the beneficial interactions between plants, animals, humans and the environment that enables community resilience, climate adaptation and local livelihoods. Vibrant exemplars of low-impact, low-carbon agriculture can be found in small-scale agroecological systems that evolved to suit local conditions. These locally developed systems are worthy of further research because they offer integrated solutions to food security, biodiversity conservation and lower-carbon futures [57]. Permaculture is a system that promotes locally developed agricultural systems that are novel or designed ecosystems using multiple species [61] including those that are centrally important to agriculture's history of innovation and transformation through the relocation and domestication of economically important species [63,64].

Domesticated and feral plant species, such as many agricultural plants and invasive weed species, enhance the novelty of ecosystems. The movement of genetic material often has unintended consequences due to many species' capacity of becoming invasive in new environments. For example, of the exotic plant species in Australia, over 25,000 species were introduced for ornamental purposes and 8000 for agricultural purposes [65]. Of these, over 3000 species have been naturalised, with 1765 species deemed to be weeds of natural environments and approximately 1200 species of agriculture [65]. Due to range expansion and deliberate or accidental introductions, new assemblages of species are producing the Anthropocene's feral futures [16]. These novel, recombinant ecosystems are rampant, forming a new nature [16]. Depending on perspectives, the functional roles of invasive plants can either be celebrated or condemned, but it is clear that different species fulfill ecological functions, such as providing habitat, sequestering carbon, stabilising stream banks or slowing erosion [16].

The way we conceive of agriculture landscapes has a distinct bearing on what we think is desirable change, through weed invasions, incremental change, designed innovations or deliberate transformations. Design-based processes that systematically iterate innovations could accelerate the adoption of integrated solutions, transforming human production and habitation systems [5], decoupling production, energy and resource use and pollution intensity [66]. However, transforming agricultural systems involves complex policy, governance and integration challenges [19] that involve local people and their politics [21]. Nevertheless, up to three-quarters of the world's terrestrial landscape is radically modified by agriculture, with huge problems [11]. Therefore, the potential of innovative, regenerative agricultural systems is immense [8,19,57,58]. Realising this potential requires the scaling up

of successful models that mobilise industries, governments and communities, in scaling-up innovations that integrate local culture, production and conservation goals, using local skills, resources and expertise [19,57] including those derived from ecological science [17] and from the experience of ecosystem restoration [37].

Furthering the theory and practice of ecological design offers prospects for systemic innovations in agroecological systems. Permaculture provides valuable ideas about ecological design that could contribute to reorientating the future of agricultural systems [67]. Permaculture systems aim to maximise productivity through actively enhancing symbiotic relationships between humans and the structural and biotic elements of gardens and farming systems. According to Mollison [61], permaculture involves the conscious design and maintenance of productive human-occupied ecosystems that mimic the diversity, stability, and resilience of ecosystems such as forests. Permaculture aims to meet human needs with minimal adverse environmental impacts through integrated landscape designs that incorporate land and water management, structures (buildings, earthworks, fencing, etc.), and people. Permaculture designs aim to use diverse species in complementary relationships to increase productivity, sustainability and multi-functionality [61,67]. Permaculture theories emphasise using diverse species to maximise productivity by optimising synergies and symbiotic relationships between multiple components—biotic and non-biotic. This approach builds on Howard Odum's theoretical work on systems ecology [67], in which he proposed designing "novel and productive ecosystems in which species are regarded as distinctive but interchangeable system components which should be selected from a global pool ... the distinctive inputs and outputs of each species will connect in novel assemblages" (Odum 1971, p. 280 as quoted in Ferguson and Lovell 2014 [67]).

Locally diverse food systems remain critical to the well-being and security of billions of people [57,68] and to the diverse biota that inhabit these landscapes. The central tenant of actively evolving novel ecosystems to meet the needs of humans, and other species, is contributing to reinventing human settlements and their agricultures [57,58,61,67]. Therefore, permaculture and agroecology deserve more support because they offer practical and theoretic guidance on how to build diverse, locally developed systems with the potential to improve the productivity of millions of villages, suburbs and shantytowns [68], while also enhancing the habitat for many other species. These ideas about integrated landscape design deserve further research and development, given the likely benefits of agroecological systems that integrate conservation and production objectives [57,61,67].

## 7. Design Ecology—Adaptively Managing Ecosystems

Design is an approach to solving problems that requires reformulating both the nature of problems and the scope of solutions. Reassembling multiple complementary components into systems that work is a generic design challenge that can manifest in many situations and at many scales [5]. Design approaches offer pathways for solving the Anthropocene's "peak everything" problem [69].

The rich history of design thinking applied to ecology includes Odum's [70] (1971) work on systems ecology and McHarg's [71] 1969 classic, *Design with Nature*. Design-based approaches provide an organising framework for mobilising ecological science into a wide range of disciplines, including engineering, architecture and urban design, land-use planning and agricultural. This idea is gaining widespread momentum under the banner of nature-based solutions [72]. These solutions require iterations of plans and their execution, evaluation and revision through processes of adaptive management that are a form of collective experimentation in governing landscapes and their common pool resources [5].

Design-based approaches are embedded in, and empower, adaptive management, with its iterative cycles of planning (designing), acting, monitoring and reviewing that enable cycles of action and revision in a world of deepening uncertainties [5]. These uncertainties are accelerating with climate change presenting fundamental challenges for integrated assessments, planning, and management of landscapes [35]. Complexities, uncertainties, multiple perspectives and high-stakes decisions are characteristics of post-

normal or "post-natural" situations [39]. Therefore, the challenges of ecological design in the Anthropocene are conceptual, theoretical and philosophical, involving the challenges of governing uncertain climatic futures and working in post-normal conditions [73]. In these post-normal conditions, "natural resource management" challenges are a result of the environment shifting away from any benchmarks or "standards" established by empiricism under stationary historical conditions [73]. Measurements or models of historical conditions no longer provide reliable templates for establishing goals for ecosystems or species conservation [73], with increasing uncertainties arising from ecosystems' capacity to shift to radically altered states [46]. Understanding the critical drivers of ecosystem change and working constructively with these processes and drivers becomes critical when dynamic risks and uncertainties increase [46]. Ecological science becomes increasingly valuable under conditions of rapid change because socio-ecological systems require governance based on paradigms capable of handling rapid, non-linear change [74–77]. Iterative, designed-based and experimental approaches offer options for solving complex ecosystem management problems under dynamic conditions [5]. Defining what constitutes the goals for the sustainable management of landscapes requires clearly defined future-oriented objectives that, while being informed by ecological science, are realistic about the nature of the dynamically changing world [49]. These objectives will need to be socially determined, through governance processes that use design-based approaches to solving complex problems [78]. Policy design is an idea that means we need to think carefully about policy interventions in a world of deepening complexity and uncertainty.

By actively embracing nature-based solutions and ecosystem design, we are in fact promoting designer ecosystems. Design-based approaches require acceptance that landscapes, and other commons, need to be managed and that this management needs to be governed. We need to collectively set explicit objectives for ecosystems and recognise this as a creative design or planning exercise, in which we are clear about the theories and planning processes used for setting objectives for ecosystems and the means—or management options—that work to achieve these ends.

## 8. Conclusions

This paper explores concepts of cultural landscapes, ecological design and designer ecosystems in an attempt to identify integrated options for managing agro-ecosystems under Anthropocene conditions. The paper argues that we need this kind of exploration for several reasons. Firstly, wilderness ideals and naturalism remain shackles that perpetuate colonial exploitation and can blind us to the cultural management and custodianship that shapes landscapes [3].

Secondly, achieving restoration ideals may be neither feasible nor desirable under Anthropocene conditions, including rapid climate change. Further, the Anthropocene is breaking down definitional boundaries, blurring the lines between what is "cultural" and "natural" [79]. Nonetheless, the humanities' global-scale problems, including the climate and biodiversity conservation crises, require urgent, creative responses [78–80] that can draw on the capabilities provided by ecology.

Finally, adopting ecological design approaches to reshaping landscapes can help solve complex sustainability problems. Designed approaches offer opportunities for meeting the needs of humans and other species, without degrading local ecosystems further or pushing the earth beyond its safe working limits [78–80]. These approaches are based in the recognition that landscapes are co-evolved assemblages of cultural and natural processes that can be steered and shaped by collective choices and actions. For this steering, we need research into theoretical models of working with dynamic, non-static world, which can be applied across multiple scales that span the local to the global, and which are implemented in ways appropriate to diverse local circumstances.

**Funding:** No funding was received for this research.

**Acknowledgments:** I acknowledge the encouragement from the editors and the work of the peer reviewers.

**Conflicts of Interest:** The author declares no conflict of interest. No funders played any role in the design of the study; in the collection, analyses, or interpretation of data; in the writing of the manuscript; or in the decision to publish the results.

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
