# Peer review of "Designer Ecosystems for the Anthropocene—Deliberately Creating Novel Ecosystems in Cultural Landscapes"

_sustainability, doi:10.3390/su14073952_

Round 1

Reviewer 1 Report

The topic is interesting and the manuscript is in general well written.

There is a clear unbalance in its structure as author redundantly focus on the outdated view of wilderness. Section 2-5 are as a whole quite long and give the impression that the same topics are repeatedly discussed while the suggestions/solutions proposed (section 6-7) are briefly and (more importantly) too generic. 

I suggest that the manuscript should be more concise in its contextualization and rationale building. It also should not focus in its justification the "purist" restoration/wilderness point of view as it is very outdated (and not used).

Finally, the author fails to explore designer ecology beyond previous publications. As one is presented to the its basis there is need to be more profound then citing agroecology as a success case (which was not created through the development of designer ecology/exosystems/systems).

Other comments

ABSTRACT

  • No proper information regarding the methodology used in the study is mentioned (citing Australia’s example is not enough).
  • No results, implications or practical suggestions are included. It’s all too generic.

L 75-77 -  I wonder whether is appropriate to consider all human-changed environments in the world as cultural-landscapes (CL). A forest subjected to clear-cutting is a cultural-landscape? If no Indigenous people actively manage that forest, I will argue that is not. Moreover, is the sole presence of humans (and therefore some management/influence/change occurs) is enough to an environment to be considered cultural-landscape? If that is the case, basically the whole world is a CL, including large tracts of Antarctica (not even high picks never climbed as they are likely to have spiritual connections, deep see regions nowadays subjected to plastic etc accumulation, or even the atmosphere). Although agriculture helped increase the scale of the human impact in the word’s ecosystems, previous human groups also exert influence in the environment and have cultural connections to them.

L83-85 – Any environment subjected to some kind of use/management has its functions and attributes (e.g. population density of a harvested tree, that also affects pollinators, fruit eaters/seed dispersers etc). Agriculture and other intense use cause larger impact and therefore more simplification.

L102-105 – Citation is needed fore this purist views. Also, it’s important to put into perspective how much this purist view of restoration indeed are important drivers.

L106-110 – It is important to elaborate on why “top-down approaches rarely deliver effective landscape management”.

L115 – Check citation.

L121-124 – As a consequence, we humans feel entitled to change/manage the whole world based on current (or as suggested in the manuscript, novel)  human-nature point of view simply because we are part of nature. To this point in the manuscript, such rationale feels simplistic and have very important ethical consequences towards other species and future (human) generations that should be addressed.

L140-156 – The problems cause by replacing local-knowledge practices is clear, the author uses an underlying view that indigenous peoples management are per se good which is not the case as there are examples of inappropriate resources management.

L126-192 – The use of Aboriginal use of land is appropriate although too limited in face of the number of different landscapes subjected to humans’ influence. Can we extend the same rationale to the Amazon? Other places? It is important to show that the same logic can be applied elsewhere.

L209-210 – It seems that restoration purists are a fringe point of view as previous paragraph describes SER’s perception change that no longer aims at going back to human-free environment. As such, the argument should not be based on a point of view that is no longer dominant but instead current practices in restoration.

L216 – and L221- I do not agree. My understanding about SER’s statement and my practical view on “ecological restoration” is to maximize, optimize or reintroduce some ecological functions onto  environments that have been degraded below acceptable levels (from land mining to overharvested forests). As such, we humans do not remove ourselves from nature but actively try to create environments with better attributes (established by humans, of course). There is no intention to “restore to original wilderness”.

L252 – I am not sure what is the meaning of socio-ecological models? It refers to models that take into account the global-scale  human impacts on nature  such climate change? Also, the change from “stationary” models to “dynamic” is largely based on the accumulation of knowledge of natural processes that has led to the improvement of modeling/models.

L266-270 – All attempts to restore (maybe and solely partially apart from “purists”) are planned and accepts humans as inducers. Authors has been quite repetitive here and using an unfounded logic.

L274-278 – Paragraph redundant.

L282-283 – It is not clear what this part has to do with the rest of the paragraph.

L286- check full stop after “with”.

L302 – 323 – “disturbance also underpin definitions of wilderness as areas relatively undisturbed by human activities” Although disturbance can indeed be put into the context of humans it by no means is restricted to that. Disturbance is also a natural process that causes heterogeneity. Author is again restricting concepts as a way to justify the manuscript’s logic.

Reviewer 2 Report

Dear author,

This review includes a critical and constructive analysis of the existing published literature in the field of research analyzed (eg cultural landscapes, ecological design and designer ecosystems).

Recommendations:

  1. Extending the bibliography in this paper. I have identified a number of 80 bibliographic titles, it seems to me quite a bit for such a type of research.
  2. The conclusions should reflect your point of view, not include citations.

Sincerely,

Author Response

I thank you for your supportive review

Round 2

Reviewer 1 Report

I am happy for the author's interest in evaluating my suggestions and incorporating them into the reviewed version of the manuscript.